# Carbonic Anhydrase IX and Survivin in Colorectal Adenocarcinoma Cells: Slovakian Population Study

**DOI:** 10.3390/biology10090872

**Published:** 2021-09-04

**Authors:** Zuzana Kováčová, Ingrid Hodorová

**Affiliations:** Department of Anatomy, Pavol Jozef Safarik University in Kosice, 041 80 Kosice, Slovakia; zuzana.kovacova@upjs.sk

**Keywords:** carbonic anhydrase IX, colorectal cancer, immunohistochemistry, survivin

## Abstract

**Simple Summary:**

This retrospective study (Slovakian population study) brings information about immunohistochemical detection of CAIX and survivin in 74 samples of human colorectal adenocarcinoma and comparison their expression with expression in healthy colon tissue. Our results show that all of samples with healthy colon tissue were CAIX and survivin-negative and there is no statistically significant dependence of these proteins and the chosen clinicopathological parameters. These findings demonstrate that detection of these proteins could be useful for tumor diagnostic and prognostic and CAIX and survivin could represent independent negative prognostic markers of colorectal cancer.

**Abstract:**

The aim of this study was to detect carbonic anhydrase IX (CAIX) and survivin in the colorectal adenocarcinoma cells of the Slovakian population. We used an indirect three-step immunohistochemical method with DAB staining for the localization of the proteins and investigation their expression. We compared their expression with expression in healthy colorectal tissue. In 74 tissues of colorectal adenocarcinomas, 42% showed CAIX positivity and 20% showed survivin positivity. Brown membrane immunostaining was visible in CAIX-positive tumors. Survivin-positive tumors had strong brown cytoplasmic immunostaining. Co-expression of both proteins was present in five specimens (7%). The samples of normal colorectal tissue (without carcinoma) were CAIX-negative and survivin-negative. We also applied the Chi-squared test for evaluation statistically significant association between the expression of proteins and selected clinical and histopathological parameters. We did not find any statistically significant correlations between CAIX or survivin expression and sex of patients, the grade of the tumor, nodal status and presence of metastasis (*p* > 0.05). The fact that all samples of normal colorectal tissue were CAIX- and survivin-negative could lead to the possibility of using these two proteins as potential tumor diagnostic markers. On the basic of the available publications and data, we suggest that CAIX and survivin could be negative independent prognostic markers of colorectal cancer, which could affect response to therapy.

## 1. Introduction

Colorectal cancer represents a global health problem, being, for example, the second most common cause of cancer death in the United States [1]. The Slovak Republic has one of the highest incidences of this disease in Europe [2]. Therefore, scientists have been searching for specific biomarkers or potential prognostic factors that could be applied in the diagnosis and treatment monitoring of patients with colon cancer or that could affect the character of a tumor cell.

Carbonic anhydrase IX (CAIX), as a transmembrane metalloprotein, catalyzes the reversible hydration of carbon dioxide to bicarbonate ions and protons [3]. The experimental and clinical evidence shows that CAIX can provide several selective advantages for tumor cells [4]. Due to the rapid proliferation of tumor cells and increased requirements of newly emerging tumor tissue for oxygen supply, the areas with inadequate vascularization form into a tumor mass [5]. Glycolytic metabolism causes the accumulation of lactic acid and carbon dioxide that contribute to a decrease in the extracellular pH. Tumor cells have to adapt to these conditions if they want to survive and CAIX plays a major role in this adaptation process [6]. The structure of CAIX contains a PG domain (proteoglycan-like region) that affects cell adhesion. The interaction of the PG domain of CAIX and catenin causes a decreased E-cadherin (epithelial cadherin)-mediated cell adhesion, and the deregulated cell–cell adhesion is a prerequisite for tumor invasion and formation of metastasis [7]. CAIX exosomes also induce expression of MMP-2 (matrix metalloproteinase 2), and MMP-2 cleaves major macromolecules of the extracellular matrix as collagen IV, which plays critical roles in cell migration, invasion of cells and metastasis formation [8,9]. Another study also showed an increased resistance of tumor cells to therapy associated with high expression of CAIX [10].

Survivin is the smallest member of the inhibitor of apoptosis (IAP) protein family and prevents apoptosis through the inhibition of effector caspase 3 and caspase 7 [11]. As a key regulator of programmed cellular death, it also affects mitosis through the regulation of the spindle mitotic checkpoint. It can be detected in the nucleus, but also in cytoplasm of tumor cells [12]. Depending on the localization in a tumor cell, its function is different too. In the cytoplasm of the cell, it works as antiapoptotic protein and nuclear survivin which probably regulates the cell division [13]. Similar to CAIX, potentiated expression of survivin can cause upregulation of MMP-7 expression (matrix metalloproteinase 7). MMP-7 plays a role in the degradation of extracellular matrix and basement membrane and this molecular mechanism is crucial for survivin-mediated invasiveness of tumor cells [14,15]. In addition to tumor invasiveness and aggressiveness, survivin probably affects tumorigenesis. Some authors have reported that survivin expression induces global transcriptional changes in the tissue microenvironment of the urinary bladder that may promote tumorigenesis, and that survivin stimulates colon adenomas with mild dysplasia into highly dysplastic lesions that play important role in colorectal tumorigenesis [16,17]. Because most types of antitumor therapy induce tumor regression through the induction of cell apoptosis, survivin also has a possible role in determining the chemo- and radio-sensitivity profiles of tumor cells [18].

For these reasons, and for character and function of CAIX and survivin, the aim of the study was to detect these two proteins in colon cancer cells as potential prognostic factors. 

## 2. Materials and Methods

### 2.1. Patients

In our work, we used 74 samples of human colorectal adenocarcinomas. Our results were compared with respect to basic clinical and histopathological parameters such as sex, tumor grade, nodal status or presence/absence of metastasis. The formalin-fixed and paraffin-embedded tissues were retrieved from archive sources of the Department of Pathology, Louis Pasteur University Hospital, Kosice, Slovak Republic. The characteristics of the study group are summarized in Table 1.

In our work we also used 5 samples of human colorectal tissue (without carcinoma) which were retrieved from archive sources of the Department of Histology, Louis Pasteur University Hospital, Kosice, Slovak Republic. 

### 2.2. Immunohistochemical Detection of CAIX and Survivin, Scoring Method and Statistical Analysis

For CAIX immunohistochemical detection, polyclonal rabbit primary antibody (bs-4029R, Bioss antibodies) was used, and for surviving immunohistochemical detection monoclonal mouse anti-human primary antibody (clone 12C4, Dako North America, Inc.) was used. After dewaxing, 3-mm sections were washed in wash buffer (Dako Agilent Pathology Solution). Then, the sections were treated by 30% H_2_O_2_ in methanol for 30 min in room temperature to reduce endogenous peroxidase activity. The process continued with blocking nonspecific staining with blocking serum (2% milk buffer containing skim milk in TRIS buffer) for 30 min at room temperature, and primary antibody was applied overnight in a humidified chamber at 4 °C. After rinsing in wash buffer, the sections were incubated in a humidified chamber with the secondary antibody (biotinylated link, Dako LSAB2 Kit) for 30 min at room temperature. Later, the slides were washed with wash buffer and subsequently incubated in a humidified chamber with streptavidin-horseradish peroxidase (HRP) label (Dako LSAB2 kit) for 30 min at room temperature. After rinsing in a wash buffer, the protein was visualized with 3.3′-diaminobenzidine tetrahydrochloride (DAB). The slides were counterstained with Mayer’s hematoxylin, then washed in tap water, dried, mounted, and cover-slipped. The immunohistochemical process was controlled by omitting primary antibody (negative control).

The results of the immunostaining were evaluated using light microscopy independently by two observers blinded to the clinicopathological parameters. We used LAS EZ (Leica Application Suite, version 2.0.0) software. The expression of CAIX and survivin was quantified using a visual grading system based on the extent of staining (percentage of positive tumor cells), graded on a scale of [−] to [+++] (0% of positive cells = minus [−]; up to 10% of positive cells = [+]; 11–90% of positive cells = [++]; 91–100% of positive cells = [+++]). As positive specimens, only [++] and [+++] samples were considered, and as negative specimens, only [−] and [+] samples were considered [19]. We used the Chi-squared test to evaluate the statistically significant association between the expression of proteins and clinical and histopathological parameters. Only *p* < 0.05 was considered significant.

## 3. Results

### 3.1. Immunohistochemical CAIX and Survivin Expression in Colorectal Adenocarcinoma Samples

CAIX-positive tumor cells show strong brown membrane staining. In 74 adenocarcinoma specimens, 31 samples were CAIX- positive and 43 CAIX-negative (Figure 1A,B). Survivin was detected only in cytoplasm, and no nuclear reaction for the survivin protein was observed in any analyzed specimens. The positive tumor cells showed a strong brown cytoplasmic staining and the number of survivin-positive samples was 15. A total of 59 tissues were survivin-negative (Figure 2A,B). Details describing CAIX and survivin expression in colorectal adenocarcinomas are illustrated in Table 2.

### 3.2. Co-Expression of CAIX and Survivin

In this study, we also monitored co-expression of both proteins in adenocarcinoma cells. A total of 33 samples (45%) were CAIX- and survivin-negative. We detected co-expression of both proteins in five specimens (7%). Different expression of CAIX and survivin was observed in 36 cases (49%).

### 3.3. Immunohistochemical CAIX and Survivin Expression in Normal Colorectal Tissue

All of the samples were CAIX-negative. No nuclear or cytoplasmic reaction for the survivin protein was observed in any of the analyzed specimens.

### 3.4. Statistical Analysis

We did not find any correlation between expression of either of the proteins and clinicopathological parameters (sex of patients, grade of tumor, nodal status and absence/presence of metastasis) by statistical analysis. We applied the Chi-squared test (available as free statistical software on Internet, www.socscistatistics.com, accessed on 18 June 2020), which revealed no statistically significant dependence (*p* > 0.05). The results of our statistical evaluation are shown in Table 3 and Table 4.

## 4. Discussion

In the present study, we investigated the expression of CAIX and survivin in adenocarcinoma colorectal cells in a Slovakian population by immunohistochemical detection. Adenocarcinomas represent more than 90 % of all histological types of colorectal tumor and originate from epithelial cells of the colorectal mucosa [20]. Epidemiological data shows that the incidence of colorectal carcinoma has been steadily rising globally, and colorectal carcinoma is the most diagnosed cancer disease among men in Slovak Republic [2]. High levels of CAIX and survivin have been identified in a wide range of malignancies [8,21]. However, the fact that CAIX is missing in the majority of normal tissues (being abundant only in the epithelium of the stomach and gallbladder) [4] and survivin expression is minimal or absent in normal tissues [22] suggests, that immunohistochemical detection of these two proteins could be useful for tumor diagnostic and prognostic.

We detected CAIX in 32 samples of 74, representing 42% of tissues which is in agreement with Saarnio et al. [23], who reported expression in 47% of colorectal adenocarcinomas. The CAIX expression was not significantly associated with the sex of patients, grade of tumor, nodal status or presence/absence of metastasis. Similar results were described by Korkeila et al. [24], who investigated CAIX expression in 166 samples of rectal carcinoma, finding that CAIX expression pattern was not significantly associated with the selected parameters, for example, the size-nodal status, or grade of the tumor. Additionally, Niemelä et al. [25] reported that the expression of this transmembrane protein was not significantly correlated with tumor differentiation grade.

Survivin was detected in 20% of cases and positive expression was found in the cell cytoplasm. These findings are consistent with observations reported in studies by Jakubowska et al. [26], who noted that the positive expression of survivin was more common in the cell cytoplasm, and Ponelle et al. [27], who also used immunohistochemistry and confirmed a higher incidence of cytoplasmic survivin expression than in the nucleus in adenocarcinoma cells of colon. Similarly, as in the case of CAIX, the Chi-squared test did not confirm significant association between survivin expression and the selected clinical and histopathological parameters. Heidari et al. [28] also reported no significant association between survivin expression and lymph-node metastasis in colorectal cancer cells and Lin et al. [29] also reported in their study that there was no correlation between the histological differentiation grade of colorectal carcinoma and the immunoreactive intensity of survivin.

The fact that all of the samples of normal colorectal tissue were CAIX- and survivin- negative could present the possibility of using these two proteins as potential tumor diagnostic markers. Our results also show that there was no significant difference in the expression of either protein with differences in the grade of colorectal adenocarcinoma, sex of patients, status of lymph nodes and presence of metastasis. This means, that the immunohistochemical detection of the CAIX and survivin in colorectal adenocarcinoma does not provide information about the clinicopathological parameters of the tumor, only about the pathological process or tumorigenesis that has already begun in colorectal tissue.

On the basic of our results and on the characteristics and function of CAIX and survivin, we suggest, that CAIX and survivin could represent independent negative prognostic markers of colorectal cancer. Kuijik et al. [30], in their review study, mentioned that patients with high CAIX expression in colorectal carcinomas had shorter disease-free survival, progression-free survival and worse metastasis-free survival. On the contrary, it was found that patients with rectal cancer and with negative or weak CAIX staining intensity had significantly longer disease-free survival in the study ofKorkeila et al. [24]. Gu et al. [31] noted that patients with survivin-positive colorectal adenocarcinomas had a shorter length of time before the disease relapsed (relapse-free survival).

Expression of these proteins may affect the effect of anti-tumor therapy. Approximately half of our samples were CAIX-positive. Therapy selection could take into account whether the tumor cell is CAIX-positive or negative. Namely, some preclinical studies have shown that cells with higher expression of CAIX have lower sensitivity to some experimental therapies (chemotherapy, radiotherapy and antiangiogenic therapy). At the same time, after inhibition of the catalytic activity of transmembrane protein CAIX, there was an increase in sensitivity to treatment [10,32].

In the case of the protein survivin, some authors have reported that prognosis of colorectal carcinoma was affected by different localization of survivin (cytoplasmic or nuclear) in the tumor cell. In our results, tissues only exhibited positive cytoplasmic expression of survivin, which, in accordance with some research, is connected with worse prognosis. For example, Qi et al. [33] reported that in 142 cases of colorectal cancer, higher cytoplasmic survivin expression was associated with a poorer prognosis, and in contrast, higher nuclear survivin expression was associated with a better prognosis. During the selection of anticancer therapy, we could consider this localization of detected survivin in tumor cells too. This means that, as we already mentioned, cytoplasmically localized surviving mainly inhibits apoptosis of tumor cells, which affects the tumor cell cycle. Therefore, we assume that tumors with higher cytoplasmic survivin expression should be treated by apoptosis-inducing anticancer drugs.

## 5. Conclusions

In this study, we used an indirect three-step immunohistochemical method with DAB staining for the detection of the proteins as CAIX and survivin in human colorectal adenocarcinomas and compared their expression with expression in healthy colon tissue. All of samples without adenocarcinoma were CAIX- and survivin-negative. Chi-squared test did not confirm significant association between CAIX/survivin expression and the chosen clinicopathological parameters, this means, that the immunohistochemical detection of the proteins in colorectal adenocarcinoma does not provide information about the clinicopathological parameters of the tumor. Based on our results and publicated information, we could suppose that detection of these proteins in colon tissue could bring information about the ongoing pathological process or tumorigenesis in tissue and could be useful for tumor diagnostic and prognostic.

However, our study has some limitations. The further research is needed with an expanded sample database in future. In particular, further clinical studies are needed to verify the prognostic significance of proteins in practice.

## Figures and Tables

**Figure 1 biology-10-00872-f001:**
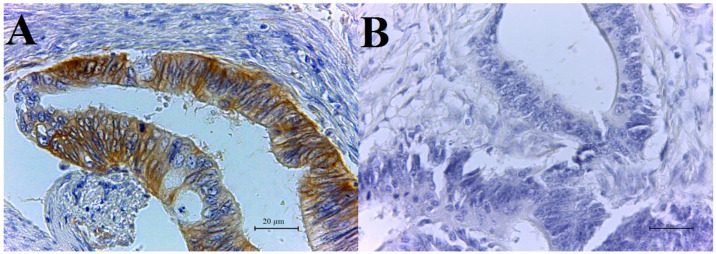
Immunohistochemical detection of CAIX with DAB staining: (**A**) CAIX-positive colorectal adenocarcinoma (high magnification), (**B**) CAIX-negative colorectal adenocarcinoma (high magnification).

**Figure 2 biology-10-00872-f002:**
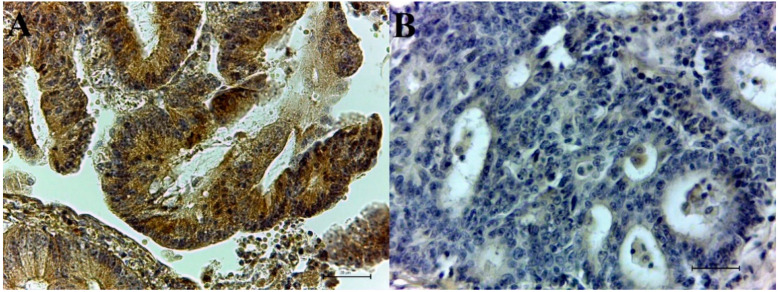
Immunohistochemical detection of survivin with DAB staining: (**A**) survivin-positive colorectal adenocarcinoma (high magnification), (**B**) survivin-negative colorectal adenocarcinoma (high magnification).

**Table 1 biology-10-00872-t001:** The characteristics of patients and tumors.

No. of Patients	*n* = 74	100%
Sex	female	30 (41%)
	male	44 (59%)
Age	≤50	2 (3%)
	>50	72 (97%)
Histological grade	G1	52 (70%)
	G2	13 (18%)
	G3	9 (12%)
Invasion to lymph node (LN)	positive LN	23 (31%)
	negative LN	51 (69%)
Metastasis	present metastasis	17 (23%)
	absent metastasis	57 (77%)

**Table 2 biology-10-00872-t002:** CAIX and survivin expression in colorectal adenocarcinomas (number and percentage of specimens).

	Quantity of CAIX Expression No./%				No./% of Negative Samples	No./% of Positive Samples
Protein	[−]	[+]	[++]	[+++]		
CAIX	27 (36%)	16 (22%)	21 (28%)	10 (14%)	43 (58%)	31 (42%)
Survivin	42 (57%)	17 (23%)	12 (16%)	3 (4%)	59 (80%)	15 (20%)

**Table 3 biology-10-00872-t003:** Statistical evaluation of CAIX expression in comparison to clinicopathological parameters (sex of patients, grade of tumor, nodal status and absence/presence of metastasis).

Clinicopathological Parameter	Quantity of CAIX Expression (No.)				No. of CAIX-Negative Samples	No. of CAIX-Positive Samples	Squared-Test
	[−]	[+]	[++]	[+++]			
Sex							*p* = 0.938
Female (*n* = 30)	10	8	7	5	18	12	
Male (*n* = 44)	17	9	13	5	26	18	
Grade of tumor							
G1 (*n* = 52)	16	10	18	8	26	26	*p* = 0.182
G2 (*n* = 13)	4	5	3	1	9	4	
G3 (*n* = 9)	7	0	1	1	7	2	
Lymph nodes (LN)							
positive LN (*n* = 23)	9	5	5	4	14	9	*p* = 0.746
negative LN (*n* = 51)	18	11	16	6	29	22	
Metastasis (MTS)							
present MTS (*n* = 17)	8	2	6	1	10	7	*p* = 0.946
absent MTS (*n* = 57)	19	14	15	9	33	24	

**Table 4 biology-10-00872-t004:** Statistical evaluation of survivin expression in comparison to clinicopathological parameters (sex of patients, grade of tumor, nodal status and absence/presence of metastasis).

Clinicopathological Parameter	Quantity of Survivin Expression (No.)				No. of Survivin-Negative Samples	No. of Survivin-Positive Samples	Chi Squared-Test
	[−]	[+]	[++]	[+++]			
Sex							*p* = 0.588
Female (*n* = 30)	15	8	7	0	23	7	
Male (*n* = 44)	27	9	5	3	36	8	
Grade of tumor							
G1 (*n* = 52)	30	13	8	1	43	9	*p* = 0.155
G2 (*n* = 13)	7	4	0	2	11	2	
G3 (*n* = 9)	5	0	4	0	5	4	
Lymph nodes (LN)							
positive LN (*n* = 23)	14	3	5	1	17	6	*p* = 0.403
negative LN (*n* = 51)	28	14	7	2	42	9	
Metastasis (MTS)							
present MTS (*n* = 17)	10	3	4	0	13	4	*p* = 0.703
absent MTS (*n* = 57)	32	14	8	3	46	11	

## Data Availability

All employed data is included in our manuscript.

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
