# Peer review of "Carbonic Anhydrase IX and Survivin in Colorectal Adenocarcinoma Cells: Slovakian Population Study"

_biology, 2021, doi:10.3390/biology10090872_

Round 1
Reviewer 1 Report
In the submitted manuscript “Carbonic Anhydrase IX and Survivin in Colorectal Adenocarcinoma Cells: Slovakian Population Study” the authors analyzed 74 tissues of colorectal adenocarcinomas. From them 42% showed CAIX positivity (31 samples were CAIX positive (only 10 were +++) and 43 CAIX negative) and 20% showed survivin positivity (15 samples, only 3 were +++).
Despite of these experimental data the authors claim that: “The fact, that there was no significant difference in the expression of both proteins among the different grade of colorectal adenocarcinoma, sex of patients, status of lymph nodes and presence of metastasis, brings possibility to use carbonic anhydrase IX and survivin as a lead target for the tumor diagnostic and prognostic and as well as for anti-cancer therapies.” So the 42% CAIX positivity and 20% survivin positivity should be a “lead target for the tumor diagnostic and prognostic and as well as for anti-cancer therapies”? Please, who could accept this? And where? In a serious department(s) of pathology? In the diagnosis of colorectal (adeno)carcinoma?
Moreover, “tumor could be diagnosed by detecting CAIX or survivin and should be treated by targeted therapy”. Please what kind of “targeted” therapy?
“Based on the characteristics and function of CAIX and survivin, we suggest that carbonic anhydrase IX and survivin could represent a negative prognostic marker of colorectal cancer.” Oh, really? Based on 74 tissue analysis with 42% CAIX positivity and 20% surviving positivity?!
I strongly do not recommend publishing this manuscript. The scientific value of it is zero, it can cause harm and confusement in the scientific literature about colorectal cancer.
Author Response
1/ Despite of these experimental data the authors claim that: “The fact, that there was no significant difference in the expression of both proteins among the different grade of colorectal adenocarcinoma, sex of patients, status of lymph nodes and presence of metastasis, brings possibility to use carbonic anhydrase IX and survivin as a lead target for the tumor diagnostic and prognostic and as well as for anti-cancer therapies.” So the 42% CAIX positivity and 20% survivin positivity should be a “lead target for the tumor diagnostic and prognostic and as well as for anti-cancer therapies”? Please, who could accept this? And where? In a serious department(s) of pathology? In the diagnosis of colorectal (adeno)carcinoma?
We apologize for this sentence, which arose from our inattention. The sentence has been removed from the manuscript.
2/ Moreover, “tumor could be diagnosed by detecting CAIX or survivin and should be treated by targeted therapy”. Please what kind of “targeted” therapy?
The word target has been removed from the manuscript.
3/ “Based on the characteristics and function of CAIX and survivin, we suggest that carbonic anhydrase IX and survivin could represent a negative prognostic marker of colorectal cancer.” Oh, really? Based on 74 tissue analysis with 42% CAIX positivity and 20% surviving positivity?!
Based on the fact that all of samples of normal colorectal tissue were CAIX and survivin negative could bring possibility to use these two proteins as potential tumor diagnostic marker.

Reviewer 2 Report
Zuzana Kováčová et al detect CAIX and Survivin expression using immunohistochemical method in the 74 colorectal adenocarcinoma cells of the Slovakian population, and found that CAIX and surviving has no significant association with the clinical feature of CRC patients.
In generally, the role of CAIX and Survivin in CRC has been reported in both clinical samples and cells experiments, no top novelty regarding the CAIX and Survivin in CRC. However, the authors limited the patients in Slovakian Population, which was the only difference compared with previous study.
Besides, there some concerns need to address:
- The author did not mention the Statistical software they used, this is important to the results, pleased added it.
- I think the comparison of CAIX and Survivin with other tumor biomarkers, such as CA199, CEA, CA125, was helpful to improve the quality of the manuscript.
- In the method part, the author mention that “As positive specimens were considered only [++] and [+++] samples and as negative specimens were determined only [-] and [+] samples.” Is any Reference provided?
- I think the statistics of CRC is too old, the newly article should be cited. Ref. “Siegel et al. Colorectal cancer statistics, 2020, CA Cancer J Clin. 2020”
- The abbreviation of Carbonic Anhydrase IX (CAIX) should be used in the whole text after the first used the full name.
Author Response
- The author did not mention the Statistical software they used, this is important to the results, pleased added it.
Information about statistical program used for calculation was added in to Material and Method section.
- I think the comparison of CAIX and Survivin with other tumor biomarkers, such as CA199, CEA, CA125, was helpful to improve the quality of the manuscript.
Thank you for your suggestion, we will include such information in our next work.
- In the method part, the author mention that “As positive specimens were considered only [++] and [+++] samples and as negative specimens were determined only [-] and [+] samples.” Is any Reference provided?
Required references were included into manuscript.
- I think the statistics of CRC is too old, the newly article should be cited. Ref. “Siegel et al. Colorectal cancer statistics, 2020, CA Cancer J Clin. 2020”
Required reference was added into manuscript.
- The abbreviation of Carbonic Anhydrase IX (CAIX) should be used in the whole text after the first used the full name.
Required abbreviations were included into manuscript.

Round 2
Reviewer 1 Report
None.
Reviewer 2 Report
The authors have revised the manuscript according to my comment.